# Burden of Traumatic Brain Injuries in Children and Adolescents in Europe: Hospital Discharges, Deaths and Years of Life Lost

**DOI:** 10.3390/children9010105

**Published:** 2022-01-13

**Authors:** Marek Majdan, Juliana Melichova, Dominika Plancikova, Patrik Sivco, Andrew I. R. Maas, Valery L. Feigin, Suzanne Polinder, Juanita A. Haagsma

**Affiliations:** 1Department of Public Health, Institute for Global Health and Epidemiology, Faculty of Health Sciences and Social Work, Trnava University, 91843 Trnava, Slovakia; juliana.melichova@truni.sk (J.M.); dominika.plancikova@truni.sk (D.P.); patrik.sivco@truni.sk (P.S.); 2Department of Neurosurgery, Antwerp University Hospital, University of Antwerp, 2000 Antwerp, Belgium; andrew.maas@uza.be; 3National Institute for Stroke and Applied Neurosciences, School of Public Health and Psychosocial Studies, Auckland University of Technology, Auckland 1142, New Zealand; valery.feigin@aut.ac.nz; 4Department of Public Health, Erasmus University Medical Center, 3015 Rotterdam, The Netherlands; s.polinder@erasmusmc.nl (S.P.); j.haagsma@erasmusmc.nl (J.A.H.)

**Keywords:** TBI, epidemiology, patient discharge, mortality, YLL

## Abstract

Children and adolescents are at high risk of traumatic brain injuries (TBI). To identify those most at risk across Europe, a comprehensive epidemiological study on the burden of TBI is needed. Our aim was to estimate the burden of TBI in the pediatric and adolescent population of Europe by calculating rates of hospital-based incidence, death and years of life lost (YLL) due to TBI in 33 countries of Europe in 2014 (most recent available data). We conducted a cross-sectional observational, population-based study. All cases with TBI in the age range 0 to 19, registered in the causes of death databases or hospital discharge databases of 33 European countries were included. Crude and age-standardized rates of hospital discharges, deaths and YLLs due to TBI; and pooled estimates for all countries combined were calculated. TBI caused 2303 deaths (71% in boys), 154,282 YLLs (68% in boys) and 441,368 hospital discharges (61% in boys) in the population of 0–19 year-olds. We estimated pooled age-standardized rates of death (2.8, 95% CI: 2.4–3.3), YLLs (184.4, 95% CI: 151.6–217.2) and hospital discharges (344.6, 95% CI: 250.3–438.9) for the analyzed countries in 2014. The population of 15–19 year-olds had the highest rates of deaths and YLLs, and the population of 0–4 year-olds had the highest rate of hospital discharges. Detailed estimates of hospital discharge, death and YLL rates based on high-quality, standardized data may be used to develop health policies, aid decision-making and plan prevention.

## 1. Introduction

Traumatic brain injuries (TBI) have been described as a substantial public health problem. TBIs can have debilitating impacts on the victims, their relatives and societies [1,2]. Annually about 50 million people are estimated to sustain a TBI worldwide, with over half of the population having one or more TBIs over their life-course [2]. In the European Union (EU), 1.5 million people are admitted to a hospital and about 57,000 people die as a consequence of a TBI [3], with on average 25 years of life being lost due to each death [4].

Children and young adults, besides the elderly, are the populations at the highest risk of sustaining a TBI. This has been shown in reports from the US (United States) [5,6], Europe [3] and New Zealand [7,8]. A surveillance summary from the US for 2013 presented an incidence rate of TBI of 1591.5 among 0–4-year-olds, 837.6 among 5–14 year-olds, and 1080.7 among 15–24 year-olds, with a national rate for all ages of 884.2 per 100,000 [6].

Great variations in incidence of pediatric TBI between studies are observed. A recent review reported a range of incidences from 12 to 486 per 100,000 [9]. Another review published in 2014 showed a wide variation in incidence rates ranging from 54 to 1652 per 100,000, as well as in variations in case definitions and the type, and subsequent limitations of data used for the calculations [10]. Despite an extensive list of published studies, differences in case definitions, studied populations, study periods and data sources hinder the possibility to truly estimate the burden of TBI in these populations and compare the burden of TBI between countries, regions and subpopulations—which is of key importance to inform prevention strategies and service planning.

One approach to tackle these problems is to use high-quality, standardized data with unified definitions from the same period of time for estimating rates or the overall burden of pediatric and adolescent TBI. In this study, we took this approach and used standardized, administratively collected data to estimate the epidemiology of TBI in children and adolescents across Europe. Our aim was to estimate the burden of TBI in the pediatric and adolescent population of Europe by calculating rates of hospital-based incidence, death and years of life lost (YLL) due to TBI in 33 countries of Europe in 2014, and to compare these by country, sex and age using data from hospital discharge reports and death certificates.

## 2. Materials and Methods

### 2.1. The Study Design, Population, and Setting

A cross-sectional, hospital-based epidemiological study aiming to analyze the epidemiological situation of TBI in children and adolescents of 33 European countries was conducted. Administratively collected data for 2014 were used (this was the most recent year with data in the desired detail available). Children and adolescents were defined as ages 0 to 19 at discharge or death.

### 2.2. Data Sources

Four data sets were used for the analyses. First, the hospital discharges dataset [11], of Eurostat [12] (the statistical office of the EU), which summarizes data on hospital discharges from European countries, was used. These data were provided as microdata, e.g., for each patient discharge, a separate record was included. The coverage of the data is population-wide by definition, but country-specific exceptions apply (Figure 1). For a subset of 27 out of the 33 selected countries, hospital discharges could be divided into day case (discharged the same day) and in-patient (spending at least one night in the hospital). The hospital discharge dataset contains information on country, patient’s age at discharge, sex, and diagnosis at discharge.

Second, a dataset of causes of deaths due to injuries was obtained from Eurostat. These data are collected and summarized according to European regulation [13]. They were also provided on a micro level, detailing the age, sex, country, external cause of injury and nature of injury (e.g., the localization and characteristics of the injury), reported using the ICD-10 classification for each primary cause of death (Appendix A), available for 30 countries (this set of countries differed from the set used to analyze hospital discharges; see details in Appendix A).

The third dataset used were population counts of countries extracted from the Eurostat population database [14]. Mid-year counts were used, broken down by age category and sex as needed.

Fourth, data on life expectancy for each country and for various age categories were used. These data were extracted from the life expectancy dataset of Eurostat [15].

**Figure 1 children-09-00105-f001:**
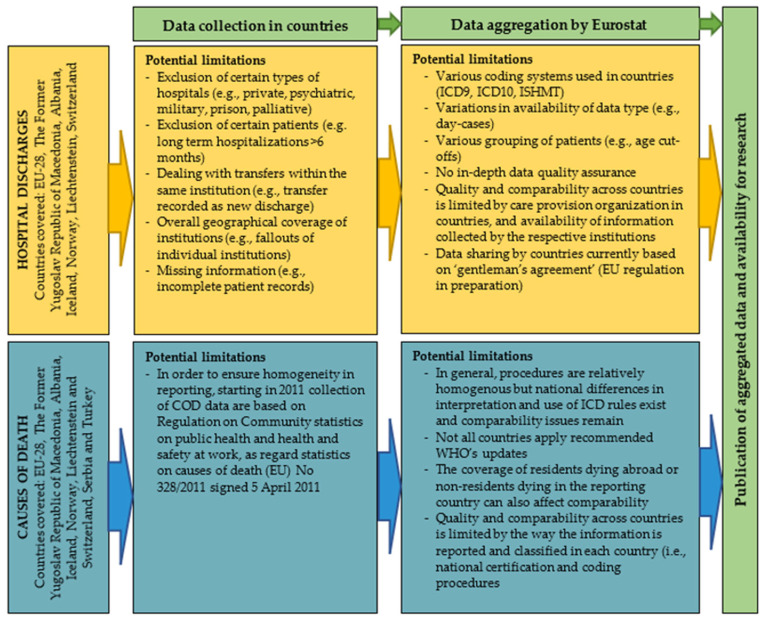
Potential causes of variation of rates of TBI-related deaths, YLLs and hospital discharges in the pediatric and adolescent populations of European countries in 2014 between the analyzed countries. Source: Eurostat hospital discharge dataset metadata [16] and Eurostat Causes of deaths dataset metadata [17]. COD: Causes of deaths, EU: European Union, ICD: International classification of diseases, WHO: World Health Organization; ICD9: International classification of diseases 9th version, ICD10: International classification of diseases 10th version, ISHMT: International shortlist for hospital morbidity tabulation.

### 2.3. Case Definition

The dataset of causes of death used the ICD-10 coding for all countries. For hospital discharges, diagnoses were coded by Eurostat using the International Shortlist for Hospital Morbidity Tabulation (ISHMT) [18]. Two diagnostic codes were used to define TBI: ISHMT 1901 (Intracranial injuries, includes ICD-10 code S06) and ISHMT 1902 (Other head injuries, includes ICD-10 codes S00–S05 and S07–S09). Appendix A provides an overview of case definitions.

### 2.4. Variable Definitions and Calculations

Three main indicators were used. First, crude hospital-based incidence rates were calculated using numbers of cases and mid-year population counts and recalculated to a population of 100,000. Age- and sex-specific rates were calculated using the following age bands: 0–4, 5–9, 10–14, 15–19 years. In the same manner, crude death rates were calculated. Third, YLLs were calculated for each death. YLLs are an epidemiological indicator that shows the sum of years that a person would have lived (based on life expectancy at the age of death) had he not died prematurely due to the consequences of TBI. These were calculated as follows:YLL = Σ d_l_ × e_l_(1)
where *d_l_* is the number of fatal cases due to health outcome *l,* and *e_l_* is the individual’s life expectancy at age of death due to health outcome *l*.

All crude rates were subsequently age-standardized using the direct method applying the European standard population defined in 2013 [19].

The pooled crude rates from our analyses were used to extrapolate our findings to the population of the EU (28 Members States before Brexit).

## 3. Results

The distribution of hospital discharges, death and YLLs across age groups and sex are shown in Table 1. Figure 2 presents a summary of age-specific rates by sex. Highest rates of hospital discharges were among those 0–4 years old, with the rest of the age groups being relatively even in both sexes. In all comparisons, boys have higher rates than girls. On the other hand, the rates of death and YLLs are highest among those 15–19 years old, followed by the 0–4 year-olds, with rates among those aged 5–14 years being lower in both boys and girls.

### 3.1. Hospital Discharges Due to TBI

Overall, 441,368 TBI-related hospital discharges were identified, of which 269,518 (61%) were boys (Table 1). The highest number of hospital discharges was among the 0–4 year-olds: 179,381 (41%). In all age groups, rates were higher in boys, and the overall sex ratio was 2.4 (Appendix A). Of all cases, 47% were intracranial injuries (ISHMT 1901), and 53% were other head injuries (ISHMT 1902), with relatively little variation across age groups (ranging from 44% among the 0–4 year-olds to 54% among the 10–14 year-olds). Appendix A shows details of this distribution per country and suggests great variation, from <10% in Malta, UK and Ireland up to 75% in Switzerland or Norway.

Crude and age-standardized rates of hospital discharges along with a pooled rate are shown in Figure 3A. The pooled age-standardized hospital discharge rate was 344.6 per 100,000 (95% CI: 250.3–438.9), with the highest rate observed for Germany (924.1; 95% CI: 919.2–929.1) and the lowest in Portugal (55.4; 95% CI: 52.3–58.8). Rates were higher among boys in all age groups and countries. The estimated age-standardized pooled hospital discharge rate was 411.0 (95% CI: 337.0–486.0) among boys and 273.6 (95% CI: 206.5–340.8) among girls (Appendix A).

The overall number of in-patient cases was 357,287 (81% of all discharges), which translated into a pooled age-standardized rate of 297.7 (95% CI: 218.1–377.3); 353.2 (95% CI: 232.7–473.7) for boys and 238.8 (95% CI: 179.9–297.7) for girls (Appendix A). The proportion of in-patients out of overall discharges did not differ significantly; it ranged from 79% among 15–19 year-olds to 85% among 10–14 year-olds.

### 3.2. Deaths Due to TBI

Overall, 2303 TBI-related deaths were identified, mostly in boys (71%). Over half of these occurred among those 15 to 19 years old (1236, 54%), followed by almost a quarter among those 0 to 4 years old (506, 22%). Table 1 presents the detailed distribution by sex and age. In general, occurrence was substantially higher in boys, most prominently among 15 to 19 year-olds with 79% boy victims. The ratio of boys vs. girls was overall 1.5 (95% CI: 1.5–1.5), ranging from 0.9 in Iceland and Malta to 4.8 in Estonia (Appendix A).

Traffic accidents were the predominant cause of fatal injury overall (63%), followed by falls and suicides (both 7%), and violence accounting for 4% of cases. In general, such distribution was observed in most countries, but relatively large differences between countries were observed. See Appendix A.

Crude and age-standardized mortality rates, along with the pooled rate for the analyzed countries, are presented in Figure 3B. A relatively large variability can be observed between the countries with age-standardized rates ranging from 1.2 (95% CI: 0.6–1.9) in Ireland to 9.0 (95% CI: 6.8–11.6) in Lithuania. The pooled rate was estimated at 2.8 per 100,000 (95% CI: 2.4–3.3) overall, at 3.9 (95% CI: 3.3–4.6) for boys—ranging from 1.4 (95% CI: 0.8–2.4) in Hungary to 11.4 (95% CI: 8.0–15.7)—and at 1.7 for girls (95% CI: 1.3–2.1)—ranging from 0.5 (95% CI: 0.1–1.6) in Ireland to 6.4 (95% CI: 3.9–10.0) in Lithuania (Appendix A).

### 3.3. Years of Life Lost Due to TBI

In total, 154,282 YLLs were estimated (68% in boys), half of these occurring among 15 to 19 year-olds (76,570, 50%), mostly in boys (59,512 YLLs, 78%) (Table 1). The overall boys-to-girls’ ratio was 2.0, ranging from 0.8 in Slovenia to 4.6 in Austria (Appendix A). Figure 3C shows the distribution of YLL rates and the pooled age-standardized rate estimated at 184.4 per 100,000 (95% CI: 151.6–217.2), with the highest rates observed in Lithuania (554.7; 95% CI: 536.6–573.3) and lowest in Ireland (78.8; 95% CI: 73.9–84.0). The pooled rates were higher in boys—245.1 (95% CI: 198.2–291.9) vs. 119.9 (95% CI: 91.8–148.0)—than in girls (Appendix A).

### 3.4. Extrapolations

Table 2 presents estimated numbers of deaths, YLLs and hospital discharges due to TBI for the EU (including the United Kingdom), calculated using the pooled value of the crude rates estimated for countries in this analysis. Based on these estimations, there were 370,951 hospital admission in the EU in 2014, of which 45% were intracranial injuries. In addition, we estimated 3087 TBI-related deaths, which lead to 199,260 YLLs. The majority of deaths (2184, 71%), YLLs (135,800, 68%) and hospital discharges (226,279, 61%) were estimated to occur in boys. See Appendix A for sex-specific details.

**Figure 3 children-09-00105-f003:**
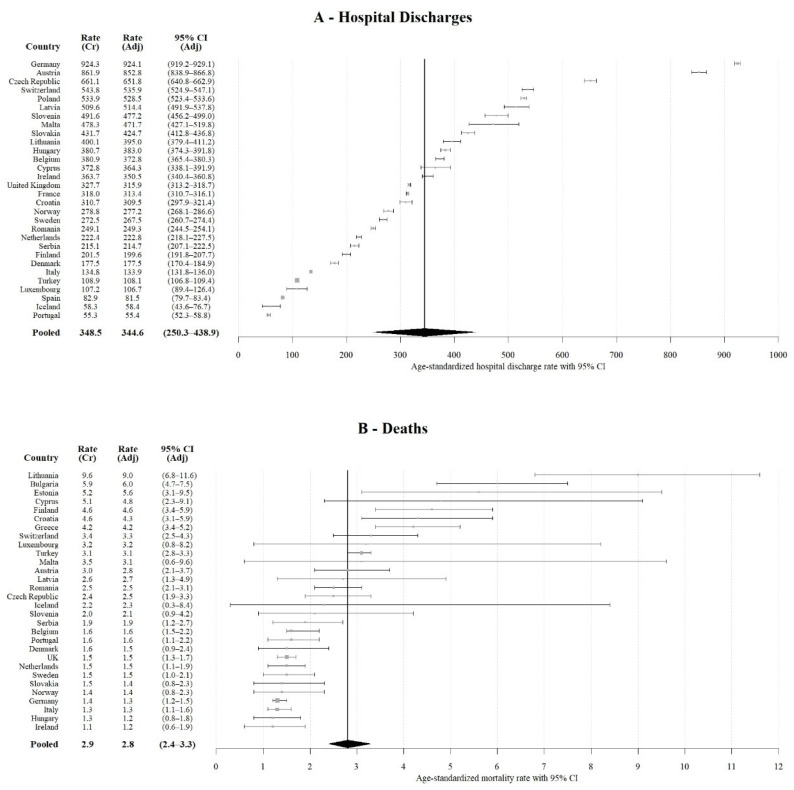
Crude and age-standardized rates of TBI-related hospital discharges (**A**), deaths (**B**) and YLLs (**C**) with estimated pooled rates for the 30 analyzed European countries in 2014.

## 4. Discussion

### 4.1. Authors’ Main Findings

We conducted a cross-sectional study in order to estimate deaths, YLLs and hospital discharges due to TBI in children and adolescents (0–19 years) in Europe. We used standardized data on hospital discharges and causes of death from 33 European countries for a single year of 2014.

We identified 441,368 hospital discharges, 2303 deaths and 105,542 YLLs due to TBI, most of them occurring in boys. We estimated pooled age-standardized rates of hospital discharges (344.6 per 100,000), death (2.8 per 100,000) and YLLs (184.4 per 100,000). We estimate that 47% of the hospital discharges were due to an intracranial injury. The population of 15–19 year-olds had the highest rates of deaths, and YLLs and the population of 0–4 year-olds had the highest rate of hospital discharges.

### 4.2. Comparison with Literature

In this study, pooled hospital discharge rates were higher, and rates of deaths were lower compared to previous European analyses in the general population.

A previous report [3] using the same data sources and case definition estimated that the pooled age-standardized rate of deaths due to TBI in the general population of Europe was 11.7 per 100,000, compared to 2.8 in children and adolescents in this study. Children and adolescents in Europe appear to be at substantially lower risk of death due to TBI than the general population. Another study produced a pooled estimate of TBI-related YLLs in the general population of Europe at 259.1 per 100,000 [4], which is about 1.4 times higher than the pooled rate of YLLs estimated in this study. Thus, even when the rate of deaths is substantially lower, the rate of YLLs due to TBI-related deaths in children and adolescents is similar to that found in the general population, suggesting similar burden of fatal cases. The lowest death rates in the 0–4-years-old group may be explained by the highest proportion of falls (32% of all deaths vs. 12% among 15–19 year-olds). On the other hand, the proportion of traffic-related injuries was lowest among 0–4 year-olds (44%), compared to older groups (66% among 5–9 year-olds, 67% among 10–14 year-olds and 69% among 15–19 year-olds). Assuming more severe injuries in traffic accidents compared to falls in the analyzed population, this may be a contributing factor to increasing mortality in older age groups.

On the other hand, the pooled age-standardized rate of hospital discharges estimated here (344.6 per 100,000) is higher than the estimate for the general population (287.2 per 100,000) [3]. This may be caused by hospitals admitting pediatric patients for observation more readily than adults. It also suggests that children and adolescents are at higher risk of sustaining a TBI requiring hospital admission than the general population. The fact that 0–4 year-olds are at the highest risk of sustaining a TBI is confirmed by other investigations: for example, a population-based study in New Zealand estimated that TBI incidence in this group in 2011 was 1300 per 100,000, which was the highest of all age groups [2]; a study from the US estimated the incidence rate among 0–4 year-olds in 2013 at 1591.5 per 100,000, which was second highest after ages 75+ [6]. In our study, the pooled age-standardized hospital discharge rate among 0–4 year-olds was 570 per 100,000; the differences between the US and New Zealand may be due to coverage of data (in the US, cases from the ER were included, and the study from New Zealand reached out even to cases not seen by a physician) or due to differences in case definitions. This suggests that our rates, in general, are underestimated and that the true incidence may be substantially higher if cases not admitted to hospitals were accounted for.

### 4.3. Implications for Research and Practice

One of the key aspects of our study is the large observed variation between the countries. These may be due to true differences, but other factors may play a role, such as variation between injury mechanisms, various coding practices in countries across Europe, different general policies pertaining to the admission of patients, inclusion criteria for care facilities or others [3,4]. This underlines the need to further standardize coding procedures across countries and to harmonize case definitions when conducting research and reporting on TBI. Based on our findings, prevention targeted to specific age groups and causes may effectively decrease the incidence and deaths due to TBI in the pediatric and adolescent population of Europe.

### 4.4. Generalizability, Limitations and Bias

Despite the fact that we have used data that were collected using standardized guidelines and for the same year, there are factors that could cause selection bias and hinder the comparability of our findings between countries, such as various data collection procedures, possible diagnosis bias due to different coding practices among countries or variations in population coverage in the analyzed countries. In addition, in some instances, TBI may not be recorded on death certificates or discharge summaries because they may have been overshadowed by other traumas or illnesses. The ISHMT 1902 coding group (other head injuries) used in this study may include some head injuries that did not result in a TBI, possibly introducing misclassification bias. We did not have the opportunity to assess the possible impact of these factors on our findings. Due to the small numbers of fatal TBIs reported in some countries, the estimated rates are more prone to error, which is reflected in wider 95% CIs. However, despite these concerns, this report is to our knowledge the most comprehensive analysis of the burden of TBI in the pediatric and adolescent populations of Europe.

## 5. Conclusions

TBIs in children and adolescents in Europe present a substantial burden that is accentuated by the young age of victims and the potential of a large number of years of lost life. This study provides estimates of death rates, YLL rates and hospital discharge rates based on standardized data that may be used to develop health policies, aid decision-making and plan prevention.

## Figures and Tables

**Figure 2 children-09-00105-f002:**
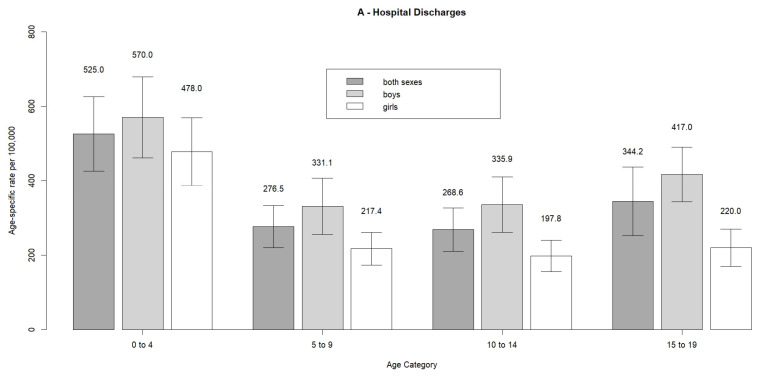
Age-standardized rates of TBI-related hospital discharges (**A**), deaths (**B**) and YLLs (**C**) with estimated pooled rates for the analyzed 30 European countries in 2014.

**Table 1 children-09-00105-t001:** Counts and relative distribution of hospital discharges, deaths and YLLs due to TBI by age and sex in 2014 in the analyzed countries.

Age Group	Hospital Discharges			Deaths			YLLs		
	Boys	Girls	Total	Boys	Girls	Total	Boys	Girls	Total
0 to 4	100,142 (56%)	79,239 (44%)	179,381	294 (58%)	212 (42%)	506	21,913 (56%)	17,039 (44%)	38,952
5 to 9	52,781 (62%)	32,330 (38%)	85,111	155 (63%)	93 (37%)	248	10,820 (61%)	6999 (39%)	17,819
10 to 14	49,678 (64%)	27,654 (36%)	77,332	205 (65%)	108 (35%)	313	13,297 (63%)	7644 (37%)	20,941
15 to 19	66,917 (67%)	32,627 (33%)	99,544	978 (79%)	258 (21%)	1236	59,512 (78%)	17,058 (22%)	76,570
Total	269,518 (61%)	171,850 (39%)	441,368	1632 (71%)	671 (29%)	2303	105,542 (68%)	48,740 (32%)	154,282

YLLs: Years of life lost.

**Table 2 children-09-00105-t002:** Estimated numbers of hospital discharges, deaths and YLLs due to TBI extrapolated to the population of 0–19 year-olds in the European Union (both sexes combined).

Variables	Pooled Crude Rate (95% CI)	Estimated Annual Number of Cases (95% CI)
Hospital discharges due to TBI–overall	348.5 (253.1–443.9)	370,951 (269,405–472,497)
Hospital discharges–intracranial injuries	156.3 (123.3–189.3)	166,369 (131,243–201,495)
Hospital discharges–other head injuries	192.2 (152.9–231.5)	204,582 (162,750–246,414)
Deaths due to TBI	2.9 (1.9–3.8)	3087 (2022–4045)
YLLs due to TBI	187.2 (153.1–221.4)	199,260 (162,963–235,663)

For extrapolation, the population estimate of 0–19 year-olds in the European Union for 2014 was used (population count 106,442,212).

## Data Availability

Publicly available datasets were analyzed in this study. This data can be found here: (https://eur-lex.europa.eu/legal-content/EN/ALL/?uri=CELEX%3A32011R0328, Accessed 10 February 2020), (https://ec.europa.eu/eurostat/cache/metadata/Annexes/hlth_act_esms_an4.pdf, Accessed 15 February 2020), (https://ec.europa.eu/eurostat/cache/metadata/en/hlth_cdeath_sims.htm, Accessed 15 February 2020),(https://www.who.int/classifications/icd/implementation/morbidity/ishmt/en/, Accessed 20 February 2020), (https://ec.europa.eu/eurostat/documents/3859598/5926869/KS-RA-13-028-EN.PDF/e713fa79-1add-44e8-b23d-5e8fa09b3f8f, Accessed 10 October 2019).

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
