# Peer review of "Burden of Traumatic Brain Injuries in Children and Adolescents in Europe: Hospital Discharges, Deaths and Years of Life Lost"

_children, 2022, doi:10.3390/children9010105_

Round 1

Reviewer 1 Report

In this article, the authors did a cross-sectional study to estimate deaths, years of life lost and hospital  discharges due to TBI in children and adolescents in Europe. They found that the population of 15–19 years old had the highest rates of deaths and years of life lost and the population of 0–4 years old had the highest rate of hospital discharges.

In addition, they explained the possible reasons for the differences between their results and others, and pointed out the limitations of their study.

The manuscript was well written, the study was soundly designed, and the results are convincing and could be very useful for helping the development of health policies.

Reviewer 2 Report

“Best of all, I pray that you will be able to do well in the COVID-19 crisis. “

Thank you for giving me the opportunity to review this wonderful manuscript. I was fascinated by the fact that this paper was analyzed based on a large database collected from several European countries.

In the manuscript, you need to use the same pattern about using the comma for every thousand units.

Line 22 Please replace ‘ranges’ with ‘range’.

Line 25 Out of context, I think you've mistyped the number of YLLs in table 1. (changed to the number of total, ‘154,282’)

Line 38-39 Your sentences, including the reference, are closer to national statistics than to the results of any research team. Please review again and substitute another reference.

<Materials and Methods>

Line 67 <2.1. Study design, population, and setting> I wonder if you have any IRB approvals that allowed you to conduct this study. I expect that even if you designed a retrospective study (with the huge database obtained from Eurostat), there would have been IRB review of it. I'm curious about your opinion on this.

Line 111 You already defined YLL in the introduction (Line 63). It doesn’t need to redefine the acronym for YLL once again. Just write ‘YLL’.
